# Genome Insights into Autopolyploid Evolution: A Case Study in *Senecio doronicum* (Asteraceae) from the Southern Alps

**DOI:** 10.3390/plants11091235

**Published:** 2022-05-02

**Authors:** Pol Fernández, Oriane Hidalgo, Ana Juan, Ilia J. Leitch, Andrew R. Leitch, Luis Palazzesi, Luca Pegoraro, Juan Viruel, Jaume Pellicer

**Affiliations:** 1Institut Botànic de Barcelona (IBB, CSIC-Ajuntament de Barcelona), Passeig del Migdia s.n., Parc de Montjuïc, 08038 Barcelona, Spain; oriane.hidalgo@ibb.csic.es; 2Royal Botanic Gardens, Kew, Kew Green, Richmond TW9 3AE, UK; i.leitch@kew.org (I.J.L.); j.viruel@kew.org (J.V.); 3Departamento de Ciencias Ambientales y Recursos Naturales, Universidad de Alicante, 03080 Alicante, Spain; ana.juan@ua.es; 4School of Biological and Chemical Sciences, Queen Mary University of London, London E1 4NS, UK; a.r.leitch@qmul.ac.uk; 5Museo Argentino de Ciencias Naturales, CONICET, División Paleobotánica, Buenos Aires C1405DJR, Argentina; lpalazzesi@macn.gov.ar; 6Biodiversity and Conservation Biology Research Unit, Swiss Federal Institute for Forest, Snow and Landscape Research WSL, 8903 Bimensdorf, Switzerland; luca.pegoraro90@gmail.com

**Keywords:** Asteraceae, cytotype, genome size, repetitive DNA, polyploidy, transposable elements

## Abstract

Polyploidy is a widespread phenomenon across angiosperms, and one of the main drivers of diversification. Whilst it frequently involves hybridisation, autopolyploidy is also an important feature of plant evolution. Minority cytotypes are frequently overlooked due to their lower frequency in populations, but the development of techniques such as flow cytometry, which enable the rapid screening of cytotype diversity across large numbers of individuals, is now providing a more comprehensive understanding of cytotype diversity within species. *Senecio doronicum* is a relatively common daisy found throughout European mountain grasslands from subalpine to almost nival elevations. We have carried out a population-level cytotype screening of 500 individuals from Tête Grosse (Alpes-de-Haute-Provence, France), confirming the coexistence of tetraploid (28.2%) and octoploid cytotypes (71.2%), but also uncovering a small number of hexaploid individuals (0.6%). The analysis of repetitive elements from short-read genome-skimming data combined with nuclear (ITS) and whole plastid DNA sequences support an autopolyploid origin of the polyploid *S. doronicum* individuals and provide molecular evidence regarding the sole contribution of tetraploids in the formation of hexaploid individuals. The evolutionary impact and resilience of the new cytotype have yet to be determined, although the coexistence of different cytotypes may indicate nascent speciation.

## 1. Introduction

Polyploidy (or whole-genome multiplication—WGM) refers to the coexistence of more than two copies of the genome in a nucleus. It arises from dysfunctional meiotic or mitotic division that results in the formation of unreduced gametes [1], generating polyploid individuals. Polyploidy is widespread across plants, and it is considered to be a major driver of evolutionary change in angiosperms [2,3]. Through the analysis of chromosome data, it is estimated that about 15% of speciation events in angiosperms involve polyploidy [4], generating similar frequencies of autopolyploid and allopolyploid species [5]. Chromosome counts and flow cytometry can efficiently identify recent polyploidisation events; however, they are limited to detect ancient polyploidisation events whose chromosomal signature might have been eroded over time by diploidising processes [6]. Nevertheless, the advent of new sequencing technologies has resulted in significant progress in our understanding of ancient polyploidy and diploidisation, revealing that WGMs have been more frequent than previously thought. For example, we currently know that the common ancestor of all angiosperms has undergone at least one ancient WGM event, predating their origin and subsequent diversification [7,8]. Since then, multiple episodes of WGM have been estimated along the major plant taxonomic lineages, supporting the role of polyploidy as one of the main engines of plant evolution [9].

Certainly, autopolyploidy results in changes in gene dosage and in levels of gene expression, which can in turn influence tolerance to environmental stress and therefore promote adaptation (e.g., [10,11,12]). Examples of autopolyploid speciation have been reported in several species, such as *Tolmiea menziesii* (Pursh) Torr. & A.Gray [13], *Heuchera micrantha* Douglas [14] and *Centaurea tentudaica* (Rivas Goday) Rivas Goday & Rivas Mart. [15] (see also a review on the topic by Parisod et al. [16] for further examples). One of the consequences of autopolyploidy is the reduction, or loss, of gene flow between ploidy levels, especially from higher to lower ploidy levels, although recurrent polyploidy formation may facilitate gene flow to higher ploidy levels. Nevertheless, the barriers to gene flow generated by polyploidy facilitate divergence within ploidy levels, increasing the potential for speciation.

Current understanding of the occurrence of autopolyploid complexes has made unprecedented progress through the application of fast and cost-effective tools such as flow cytometry, which enables reliable screening of hundreds of individuals (e.g., [17,18,19,20]) in a relatively short period of time. Indeed, large-scale analyses have facilitated the discovery of hidden minority cytotypes (e.g., *Elymus* L. [21]), which might have otherwise remained hidden using solely chromosome-based approaches. In parallel, High-Throughput Sequencing (HTS) technologies have contributed significantly to our understanding of polyploid evolution, polyploid complexes and to polyploid genome dynamics [22,23]. For example, repetitive DNA sequences, including transposable elements (TE) and tandemly arranged elements (e.g., satellite DNA), which form a substantial component of plant genomes [24,25], have been shown to diverge independently in polyploid lineages compared with their diploid progenitors [26,27]. Repeat dynamics is not only of interest to study genome organisation, function and evolution [28,29], but is shown to harbour phylogenetic signal among closely related species, based on their abundance and sequence similarity recovered from HTS analysis [30,31]. The question now arises as to whether intraspecific (and intrapopulation) repeat variation can also be identified, and how is it influenced by mechanisms such as polyploidy.

*Senecio doronicum* L. (Asteraceae) is a relatively common species of herbaceous perennial found throughout European mountain ranges between 1000–2400 m of elevation, exceptionally up to 3000 m [32]. Its habitat spans from alpine meadows to rocky screes (Figure 1). Even though diploids (2n = 20) have been described in the genus, chromosome numbers for the species have been frequently reported as 2n = 40, 80 (i.e., 2n = 4x and 8x). Additionally, higher ploidy levels (e.g., 12x) have also been found anecdotally on gametophytic records according to the Chromosome Counts Database (CCD) [33]. Aiming at evaluating and testing the existence of minority cytotypes in this species, we conducted a cytotype-screening analysis focusing on a population of the southern Alps (Tête Grosse, Alpes-de-Haute-Provence, France, Figure 2A). This analysis was combined with genome skimming approaches to characterise repetitive DNA across multiple individuals and cytotypes. Specifically, our goals were to assess: (i) the impact of polyploidy on the composition of the repetitive fraction of the genome in *S. doronicum*, (ii) investigate if there is intra- and intercytotype variation in the repetitive DNA content, and (iii) determine to what extent TEs can be used to unravel evolutionary pathways leading to polyploid complexes at the population level, beyond what can be interpreted from traditional nuclear and chloroplast markers.

## 2. Results

### 2.1. Cytotype Screening of Senecio doronicum Identifies Three Ploidies in Tête Grosse

Our flow-cytometry DNA ploidy screening confirmed the coexistence of tetraploid and octoploid cytotypes in the population (Appendix A), with the latter being over 2.5x more abundant (i.e., 141 tetraploid individuals versus 356 octoploid individuals), and with very little overlap in the distribution of each cytotype across the population (Figure 2B,C). Additionally, the analyses uncovered three hexaploid individuals growing close together (<5 cm apart), which displayed intermediate DNA ratios (i.e., genome sizes) compared to those of tetraploids and octoploids (Figure 2C). Genome sizes (Mb/1C) estimated for each ploidy level were as follows: 4x (4205.4 ± 19.56), 6x (6357 ± 29.34), 8x (8097.84 ± 48.9). The survey conducted on 36 accessions from outside Tête Grosse, in neighbouring mountain valleys, also reported a high incidence of octoploid individuals (32 out of 36; Appendix A).

### 2.2. Repetitive DNA Content in S. doronicum

A representative summary of the results of one individual per ploidy level is illustrated in Figure 3 and given in Table 1. The proportion of the genome containing repetitive elements was almost identical for the three cytotypes (i.e., mean: 85.54% (±0.32%)). Around 19–20% of the repetitive genome were unclassified elements. Ty3/Gypsy-like Long Terminal Repeats (LTR) elements dominate the repetitive landscape of each ploidy level, with genomic proportions (GP) of 46.86% in 4x, 42.02% in 8x and 44.06% in 6x, followed by Ty1/Copia LTRs (see Table 1 for detailed composition of the different repeat lineages comprising these LTR repeat superfamilies). Among four lineages of Ty3/Gypsy LTR elements identified, the chromovirus-type Tekay element was by far the most abundant (ranging between 39.08% to 43.85%). The Ty1/Copia superfamily was represented by seven lineages, with SIRE elements being the most abundant (GP: 13.08–13.60%, Table 1).

Non-LTR retrotransposons were only represented by Pararetrovirus repeats, occurring in very low abundance (i.e., ≤0.22% GP). DNA transposons included Mutator, haT and Harbinger lineages, with Mutator being the most prevalent in all three cytotype accessions (GP: 0.34–0.49%, Table 1). (N.B. Details on the number of reads analysed for the 12 individuals comprising five tetraploids, three hexaploids and four octoploids and the classification and GP of the highly repetitive elements identified in their genomes are available in Appendix A). Despite some variation observed in the GP of DNA repeats between the individuals analysed (Appendix A), we did not find significant differences in the composition of the repetitive genomes between the three cytotypes (permanova *p* = 0.41). Moreover, linear regressions of the number of reads of each repeat type between all cytotype combinations, had an R^2^ around 0.99, with a high level of statistical significance (Figure 4 and Table 2, see also Appendix A for regressions focusing on Ty1/Copia and Ty3/Gypsy elements).

The variation in TE composition between individuals with the same cytotype were also small; in all cases differences were not significant (paired Wilcoxon tests *p* = 0.3–0.89) (Appendix A), confirming high levels of similarity in the repetitive DNA content regardless of the ploidy level. The same trend was observed when comparing samples from Tête Grosse population with those analysed from outside the population (Appendix A).

### 2.3. Phylogenetic Implications of Cytotype Diversity in S. doronicum

To confirm that all recovered cytotypes belong to the same species, and to discard any individuals which showed evidence of hybridisation with closely related species, we conducted a phylogenetic analysis using the internal transcribed spacer (ITS) on a dataset comprising all 4x, 6x and 8x individuals analysed here and the extended sampling of the European clade of *Senecio* section *Crociseris* (Rchb.) Boiss. by Calvo et al. [34]. Both Neighbour-Net (NN) and Neighbour-Joining (NJ) trees confirmed that all analysed samples fell within *S. doronicum* (Figure 5A,B), but were embedded in different subclades, supporting an autopolyploid scenario, whilst indicating that octoploids most likely arose from tetraploids outside Tête Grosse. In contrast, we found that tetraploid and hexaploid individuals were grouped together in the same subclade, mixed with closely related accessions from the Alpes Maritimes. Octoploid individuals were clustered together in a separate subclade with accessions of *Jacobaea*
*kirghisica* (DC.) E.Wiebe and *S. doronicum* from other populations of the Alps and Jura. The splitstree NN also supported these relationships between Tête Grosse individuals (i.e., octoploids belonging to a separate clade from the tetraploids and hexaploids, Figure 5B). Additionally, nuclear evidence was complemented by the analysis of conserved regions in a satellite DNA repeat (207 bp) shared between all analysed individuals from Tête Grosse. From the alignment matrix, we found that all tetra- and hexaploid individuals shared exactly the same sequence while octoploids shared five unique SNPs.

Plastid DNA reconstruction was carried out on all 12 individuals, generating plastid genome sequences ranging between 151,196 and 151,222 bp. The comparison of plastid sequences between tetraploid and hexaploid individuals from Tête Grosse revealed 0–15 SNPs between the eight individuals analysed, with individual TG370 being the most divergent, although it still grouped with the other seven individuals analysed. In contrast, among octoploid individuals, 14–30 SNPs were found. Further, when comparing the plastid sequences of hexaploids and tetraploids, to octoploids from Tête Grosse, 66–87 variants were found. The plastid sequences of individuals outside Tête Grosse [FR626 and FR627 (4x), FR475 (8x)] were more similar to each other, regardless of ploidy level, than to the individuals of the same cytotype from the Tête Grosse site (Figure 5C). The NN reconstruction also supported the very close relationship between tetraploid and hexaploid individuals from Tête Grosse, while octoploids from this population appeared clustered in a different lineage (Figure 5C). Phylogenetic analyses carried out using TE outputs from RepeatExplorer2 failed to produce any systematic signal at the population level (data not shown), given the high similarity of the repetitive DNA elements between individuals and ploidy levels.

## 3. Discussion

### 3.1. Flow Cytometry Uncovers the Existence of Minority Cytotypes in S. doronicum

Our flow-cytometry-based cytotype screening not only confirmed the coexistence of tetraploids and octoploids in Tête Grosse, but also uncovered the existence of a novel minority hexaploid cytotype, illustrating the utility of this kind of approach in detecting rare cytotypes. Both auto- and allopolyploid speciation in the genus *Senecio* has been frequently reported, and the consequences at chromosome and gene levels investigated (see review [35]). Whilst diploid species are reported to occur in the genus, *S. doronicum* has so far only been reported to exist at higher ploidy levels [33]. To a large extent, and based on our field observations across the southwestern Alps, the results reported here in Tête Grosse mimic the dynamics observed more broadly for the species, where octoploids are more abundant and widely distributed than tetraploids (Figure 2A,B). Indeed, 70% of the individuals surveyed in the Tête Grosse population were octoploids. Each cytotype showed evidence of distinct habitat preference in the population studied (Figure 1), which is illustrated by the little overlap in distribution of tetraploids and octoploids across the site (Figure 2B). Whether tetraploids are, in general, being outcompeted by octoploids is still an open question, and requires future extensive samplings across the Alps. One possible explanation for such a scenario could be that octoploids have more efficient dispersal mechanisms than tetraploids. Increased rates of self-pollination and efficient local dispersal of polyploid *Ranunculus adoneus* A. Gray have been key for increased persistence and long-term expansion compared to diploids [36]. Despite both cytotypes being reported to reproduce sexually by seeds [37], clonal reproduction cannot be ruled out given the high frequency of individuals that we observed growing together (in clumps). Based on such observations, it is possible that differential rates of sexual versus asexual reproduction between cytotypes may be an important factor contributing to the contrasting evolutionary successes of the cytotypes, which in the long term may contribute to the higher frequency of the octoploids over tetraploids in the genus.

To our knowledge, the summit of Tête Grosse is one of the very few sites where both ploidy levels coexist, with other populations consisting mostly of scattered and isolated tetraploid individuals among octoploids (Figure 2A). The higher abundance of tetraploids at Tête Grosse means that opportunities for inter-cytotype hybridisation are more likely. Indeed, we observed that despite each cytotype having relatively different flowering periods, there was some overlap (c. 10 days) in flowering time between them (Pegoraro et al., unpublished), thus a window of opportunity for potential pollen exchange between cytotypes exists. Based on these observations, two main scenarios for the rise of the hexaploid individuals in the population can be considered, including a single cytotype origin or intercytotype crossing (discussed below). The very low number of hexaploid individuals found (3 out of 500 in the population) could represent the early stages of the establishment of hexaploids, but future surveys will be needed to confirm whether this is the case or just an isolated event, and potentially an evolutionary dead end.

### 3.2. Correlation of TE Amounts among Ploidy Levels Supports the Autopolyploid Evolution in S. doronicum

Among the repeat clusters identified using RepeatExplorer2, the majority were classified as Ty3/Gypsy and Ty1/Copia superfamilies, supporting the dominance of LTR retrotransposons across land-plant genomes. An overall dominance of Ty3/Gypsy-like elements has been reported in many plant genomes, including species of Asteraceae [38], but there are exceptions such as in *Urospermum* Scop. (in a different tribe from *Senecio*), where Ty1/Copia elements are the major contributors [39]. In *S. doronicum*, by far the most abundant repeats were Ty3/Gypsy-Tekay elements, with a GP of c. 43%. Indeed, these repeats comprised more than double the GP of the second most abundant repeat lineage which was the Ty1/Copia SIRE elements (GP c. 13%, Table 1). Such analyses highlight the important role of Tekay repeats in shaping the genome of *S. doronicum*. At the population scale, the repetitive landscape of the individuals analysed was highly conserved, both within cytotypes and between accessions of different ploidy levels (Figure 3, Appendix A). This was evidenced by the significant correlations observed, either when ‘all repeats’ were analysed (Figure 4, Table 2) or when the Ty3/Gypsy and Ty1/Copia repeat classes were analysed separately (Table 2, Appendix A). Together, these data provide support for an autopolyploid origin of both hexa- and octoploid *S. doronicum*.

One of the many outcomes of autopolyploidy is an increase in the amount of TEs due to the increased number of genome copies within the cell. However, mechanisms driving TE dynamics following WGM and their impact at evolutionary scales are diverse, and vary between auto- and allopolyploids and between species (reviewed in [16,40]). For example, whilst here we observed the above-mentioned significant—and almost linear—correlation between DNA repeats across ploidy levels, this is not necessarily the case in other plant species following autopolyploidy (although studies are limited). For example, a study of 300 autopolyploid *Arabidopsis arenosa* (L.) Lawalrée showed evidence of differential accumulation of TEs in different individuals, potentially influencing patters of gene expression and providing opportunities for local adaptations [41]. However, no such changes in repeat dynamics associated with cytotype was observed in *S. doronicum*, perhaps because they are recently formed cytotype variants. Although most studies of how repeat dynamics change in response to WGM come from the study of allopolyploids [40], which have increased complexity due to the associated interspecific hybridisation, the data are highlighting that genome upsizing and downsizing associated with changes in repetitive DNA activity can change over time [41]. These results therefore emphasise the importance of considering how the tempo of polyploidisation will impact the overall composition and organisation of a polyploid genome, rather than the actual polyploidisation process itself [42]. Evidence from population-wide comparative approaches is now needed to more fully evaluate the changes in the composition and organisation of repetitive elements following polyploidy over time, particularly in autopolyploids, which are very limited.

### 3.3. Unreduced Gamete Formation in Tetraploid Individuals as the Most Likely Origin of the Hexaploid Cytotype

As indicated above, two potential yet exclusive evolutionary scenarios can be invoked to explain the origin of hexaploid individuals of *S. doronicum* in Tête Grosse: Scenario 1-they have arisen from an inter-cytotype cross involving reduced gametes from both the tetraploid and octoploid, and Scenario 2-their origin solely involves tetraploid individuals, arising from a cross between reduced and unreduced gametes (which have been reported in CCD). To distinguish between these two pathways we note the following: (i) The NN splitstree analysis of the whole plastid sequence data (Figure 5C) and the SNP counting from both the satellite and plastid DNA suggest an extremely close relationship between tetraploid and hexaploid individuals; (ii) Bearing in mind that the plastid genome is maternally inherited without meiosis [43], we assume that at least, the maternal contribution to the hexaploids came from the original tetraploid gene pool at Tête Grosse; (iii) The nuclear data (ITS) also support a close relationship between tetraploids and hexaploids based on the reduced number of SNPs from the satellite analysis, and from the phylogenetic inferences depicted in Figure 5A where no ITS variants were found in hexaploids. In summary, and taking all these observations together, we propose that hexaploid individuals on Tete Grôsse most likely arose via Scenario 2. Given the small number of SNPs, together with the low number of hexaploid individuals found in this population (just three out of 500 analysed), the data also suggest that these hexaploid individuals may have arisen very recently.

Many of the examples of autopolyploid evolution reported in the literature involve diploid individuals that produce unreduced gametes leading to polyploid cytotypes (see review [16]). In contrast, our analysis suggests an origin arising from the tetraploid level. Minority cytotypes must overcome competition disadvantage and stochastic effects prior to becoming established, a process that can be favoured if recurrent WGM takes place. Certainly, the very low number of hexaploid individuals observed suggest that their long-term survival may be limited unless the production of unreduced gametes is high and/or there is the potential for vegetative growth or reproduction via apomictic pathways [44], although the latter does not seem to be present in the species [37]. In contrast, phylogenetic analyses (Figure 5A–C) indicate that octoploids have arisen from different populations of *S. doronicum*. Thus, their occurrence in sympatry suggests that long-distance dispersal has brought the populations together. The linear boundary between tetraploids and octoploids at Tête Grosse (Figure 2B) suggests expansion of one cytotype, perhaps to the detriment of the other.

## 4. Materials and Methods

### 4.1. Sampling of Senecio doronicum

For this study, we selected a population site in Tête Grosse (Alpes-de Haute-Provence, France), at an elevation of about 2000 m, where preliminary analysis of individuals had previously identified the coexistence of both tetraploid and octoploid cytotypes. Sampling of 500 individuals from this site was carried out. Thirty-six specimens from outside the population in neighbouring mountain valleys (Figure 2A) were also collected. Fresh leaf samples of 536 individuals were collected and stored in Ziploc bags at 4 °C until processed for cytotype screening using flow cytometry. Representative herbarium vouchers of each cytotype were prepared and are deposited in Royal Botanic Gardens, Kew (K).

### 4.2. Cytotype Screening by Flow Cytometry

Prior to our cytotype screening, initial analyses on individual plants involving both chromosome counts and estimations of nuclear DNA contents were carried out to enable subsequent DNA estimates to be used directly to infer DNA ploidy levels. The relative fluorescence intensities of nuclei from leaf samples were estimated by propidium iodide flow cytometry following the protocol described in Pellicer et al. [17] using a CyFlow Space flow cytometer (Sysmex-Partec, Norderstedt, Germany), fitted with a 100-mW green solid-state laser (Cobolt Samba, Solna, Sweden). Pools of five individuals were processed together. Each sample was reanalysed separately when cases of mixed-ploidy samples were detected, or in the presence of undetermined fluorescence peaks. We used *Petroselinum crispum* (Mill.) Fuss. ‘Champion Moss Curled’ [45] as calibration standard (4.5 pg/2C), and the general-purpose isolation buffer by Loureiro et al. [46] supplemented with 3% PVP-40 [47]. Resulting output histograms were analysed using the FlowMax software (v. 2.9, Sysmex-Partec GmbH, Norderstedt, Germany) for statistical calculations.

### 4.3. Genomic DNA Extraction and Illumina Sequencing

Based on the ploidy screening, five tetraploid [(TG370, TG422, TG444–Tête Grosse), (FR626, FR627-Allos)], four octoploid [(TG84, TG137, TG237–Tête Grosse), (FR475-Orciéres)] and three hexaploid individuals (TG337R, TG506, TG507–Tête Grosse) were selected for HTS analyses. Genomic DNA extraction was carried out following the CTAB method with minor modifications [48]. Samples were further processed with NucleoSpin column cleanup following the manufacturer’s protocol. DNA products were run on an agarose 1% gel and quality-control-assessed using a Qubit 3 fluorometer (Thermo Fisher Scientific, Waltham, MA, USA). NEBNext® UltraTM II DNA Library Prep Kit for Illumina® (New England Biolabs, Ipswich, MA, USA) with an average insert size of 350–500 bp were prepared and sequenced on a MiSeq v.3 platform (Illumina, San Diego, CA, USA), generating 150 nt paired-end reads (0.45–1 × genome coverage) at Queen Mary University of London Genome Centre.

### 4.4. Flow-Graph-Based Clustering in RepeatExplorer2

Raw Illumina reads were inspected using FASTQC v.0.11.9, available through the web (https://www.bioinformatics.babraham.ac.uk/projects/fastqc/, accessed on 10 January 2021), to check for low-quality reads or adapter sequences. Trimmomatic v.0.39 [49] was used to trim low-quality bases and reads were reduced to a minimum length of 100 nt (settings: AVGQUAL:20 MINLEN:100 LEADING:20 TRAILING:20 SLIDINGWINDOW:4:20). A map to reference to the complete *Senecio vulgaris* L. (GenBank Acc.: MH746728.1) plastid sequence was performed using Geneious Prime v.2021.2.2, and matching reads were excluded from further analyses. Paired reads were analysed using the RepeatExplorer2 pipeline [50,51], which performs the classification and quantification of repetitive elements using the REXdb database, which includes all known repeat elements in plants [52]. First, a preliminary analysis with one individual of each ploidy (i.e., 4x, 6x and 8x) was run to evaluate the number of reads that the server could analyse due to memory limitations. Next, a clustering analysis was performed for each of the individuals using the same number of reads, based on the maximum number of accepted reads (1.2 M reads), as advised by the pipeline developers. The annotation of the different clusters from the output directories for each individual was manually revised.

Prior to performing comparative analyses, the results of the individual clustering within each ploidy level were compared to evaluate if there were significant differences in TE composition using paired Wilcoxon tests (calculated with *wilcox.test* function in R [53]). Since no significant differences were found between individuals of the same ploidy (*p* > 0.05), one individual of each ploidy was randomly chosen for the comparative analysis (i.e., 4x: TG422, 6x: TG507, 8x: TG84). Summary tables and a barplot showing the composition of the repetitive genome in the three individuals was constructed using ggplot2 [54] in R. The comparative analysis in RepeatExplorer2 included proportional amounts of reads of each individual according to their ploidy level (4x: 0.6 M, 6x: 0.9 M and 8x: 1.2 M reads). The annotation of repeats was carried out as described above. Comparison of the proportion of different types of repetitive elements was carried out by plotting pairwise scatterplots comparing the number of shared reads of each DNA repeat class between the different cytotypes as in Pellicer et al. [55], where the slope of the plot represents the genome size ratio between cytotypes. Linear regressions were performed using the *lm* function in R and differences between genome compositions were checked by performing a permanova test using the *adonis* function in the vegan package in R.

Finally, the three most abundant shared nuclear satellites (identified by RepeatExplorer2) were checked for conserved domains. Cleaned reads were mapped to the three satellites using a map to reference approach in Geneious Prime v.2021.2.2. A single conserved region was identified at a 90% identity threshold for the three cytotypes in the most abundant satellite, and this region was reconstructed using a map-to-reference approach for all individuals in Tête Grosse. A matrix with all 12 individuals was created and aligned with MAFFT v.7.450 [56].

### 4.5. Phylogenetic Analyses and Plastid Reconstruction

The nuclear region of *Senecio* specimens belonging to *Senecio* section *Crociseris* containing the ITS1, 5.8S and ITS2 sequences in the 45S ribosomal DNA unit from the study by Calvo et al. [34] were downloaded from GenBank. To recover the same region from our individuals, the RepeatExplorer2 outputs annotated as ribosomal DNA clusters were identified, and the largest contig produced was retrieved. We double-checked the validity of these contigs by mapping all reads at a 90% similarity threshold and we did not find variants. These sequences were added to the matrix and aligned using MAFFT v.7.450. The alignment was manually trimmed and inspected for inconsistencies. The resulting nexus file was analysed with Splitstree v.4.17.1 under a Neighbour-Net (NN) approach with a bootstrap of 10,000 replicates [57]. Additionally, a Neighbour Joining (NJ) tree was produced in Geneious Prime v.2021.2.2 assigning *Senecio umbrosus* Waldst. & Kit. as the outgroup based on the results of the above-mentioned study.

For the plastid reconstruction, adapter-free reads were analysed with NOVOPlasty v.4.3.1 [58] using default parameters. Several contigs were retrieved from each individual, which were mapped to a complete plastid sequence of *Senecio vulgaris* in Geneious Prime v.2021.2.2. The contigs covered 100% of the reference and a consensus for each individual was retrieved. Moreover, the cleaned and paired reads with Trimmomatic v.0.39 [49] were then mapped to the same *Senecio vulgaris* reference in Geneious Prime v.2021.2.2. From this analysis a consensus1 with a similarity threshold of 90% was extracted. In addition, a *de novo* assembly with SPAdes v.3.15.2 [59] was performed using the cleaned reads, producing several scaffolds. Those were mapped to consensus1 to check for possible insertions/deletions that could have been missed and a consensus2 was extracted. We aligned the consensus2 of each individual with the consensus obtained with the NOVOplasty method and found no differences. From the complete plastid sequences, a phylogenetic tree was produced using Splitstree v.4.17.1 [60] under a Neighbour-Net approach with a bootstrap of 10,000 replicates. Finally, following the methodology described in Vitales et al. [30], we explored phylogenetic signal in the most abundant TEs of our dataset. For that, we ran a test in which we used the top 25 most abundant repetitive elements from each cytotype.

## 5. Conclusions

This study evidences how population surveys can be helpful to uncover hidden genomic diversity, in this case illustrated by minority cytotypes, which provide fundamental information necessary to interpret the evolutionary implications of polyploidy in plant evolution. In addition, despite the limitations of using TEs as phylogenetic markers at the population level, we find compelling evidence for an autopolyploid origin of hexaploid and octoploid *S. doronicum* based on TE content. This is further supported through the analysis of nuclear and plastid markers. Future work involving hypervariable markers across populations, such as nuclear microsatellite analysis or RAD sequencing, will be an ideal complement to confirm the results presented in this work. In addition, insights into the activity of the repetitive sequences through analysis of transcriptome and epigenome data will further enhance our understanding of how autopolyploidy impacts genome dynamics and contributes to their long-term survival.

## Figures and Tables

**Figure 1 plants-11-01235-f001:**
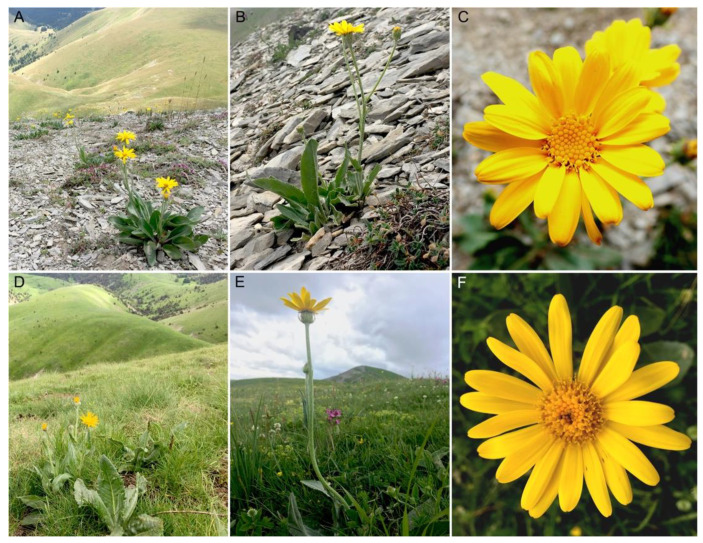
Habitat preferences and morphology of tetraploid (**A**–**C**) and octoploid (**D**–**F**) cytotypes observed in *Senecio doronicum* from the population of Tête Grosse (Alpes-de-Haute-Provence, France).

**Figure 2 plants-11-01235-f002:**
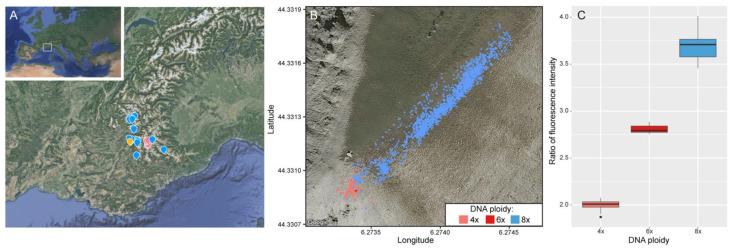
(**A**) Geographical distribution of the population of *Senecio doronicum* from Tête Grosse (
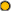
). Additional sites sampled in the Southwestern French Alps are also indicated in the map and the cytotypes recovered indicated (4x = 
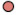
, 8x = 
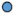
). (**B**) Distribution of individuals in the population of Tête Grosse, colored according to their cytotype. (**C**) Boxplots depicting the DNA ploidy levels assigned on the basis of relative fluorescence ratios of nuclei. Data Maps from Google Earth: Google Landsat/Copernicus Data SIO, NOAA, U.S. Navy, NGA, GEBCO. GeoBasis-DE/BKG (©2009) Inst. Geogr. Nacional.

**Figure 3 plants-11-01235-f003:**
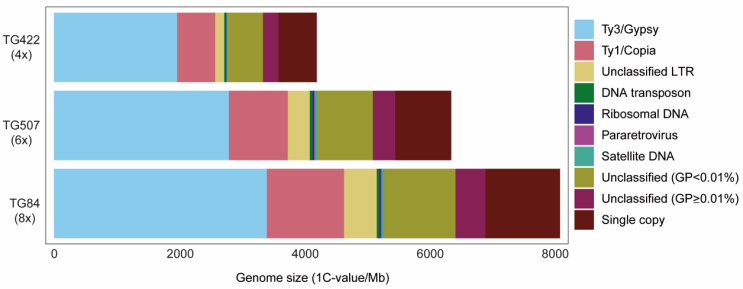
Genomic composition of *Senecio doronicum* representative of each cytotype (4x, 6x and 8x). Estimates of the genomic abundances (in Mb/1C) of different repeats are indicated and colored by repeat class [GP = Genome proportion (%), LTR = Long Terminal Repeat].

**Figure 4 plants-11-01235-f004:**
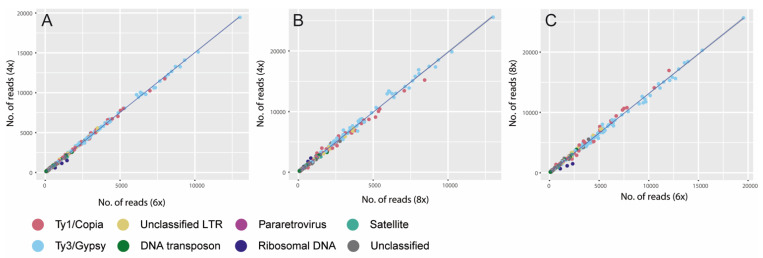
Pairwise scatterplot comparisons of the number of reads included in repeat clusters from each cytotype. The slope indicates the genome size ratio between each cytotype. (**A**) 4x vs. 6x. (**B**) 4x vs. 8x. (**C**) 8x vs. 6x.

**Figure 5 plants-11-01235-f005:**
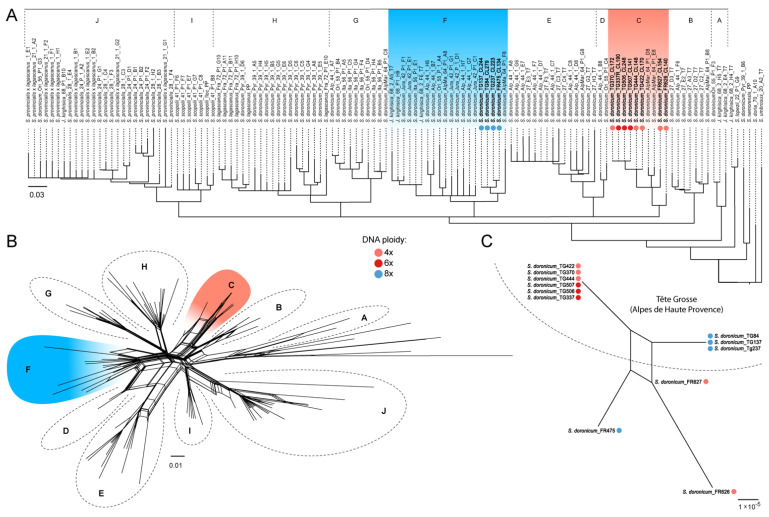
Neighbour joining tree (**A**) and Neighbour net analysis (**B**) based on ITS sequences of European clade of *Senecio* sect. *Crociseris* from Calvo et al. [34], and including 4x, 6x and 8x individuals of *Senecio doronicum* from the present study. (**C**) Neighbour net analysis of whole plastid sequences among those same individuals sequenced in our study. Dashed line separates individuals of Tête Grosse from other populations in the area. Capital letters indicate genetic clusters.

**Table 1 plants-11-01235-t001:** Repetitive DNA composition estimated in individuals TG422 (4x), TG507 (6x) and TG84 (8x) as illustrative of the overall dynamics reported across cytotypes.

		Genome Proportion (GP)					
		4x		6x		8x	
**Repeat Type**	**Lineage**	**[%]**	**[Mb]**	**[%]**	**[Mb]**	**[%]**	**[Mb]**
**Ty1/Copia**		14.51	610.40	14.86	944.78	15.28	1237.56
	SIRE	13.08	550.07	13.34	848.01	13.60	1101.70
	Angela	0.97	40.96	1.11	70.64	0.95	76.83
	TAR	0.11	4.64	0.10	6.43	0.14	11.51
	Bianca	0.06	2.44	0.06	3.99	0.31	24.73
	Ale	0.03	1.24	0.01	0.82	0.03	2.81
	Tork	0.06	2.37	0.04	2.48	0.06	4.60
	Ikeros	0.21	8.68	0.20	12.42	0.19	15.39
**Ty3/Gypsy**		46.86	1970.85	44.02	2798.58	42.06	3406.09
	Tekay	43.85	1843.88	40.61	2581.82	39.08	3164.91
	Athila	1.67	70.09	1.60	101.55	1.56	126.28
	CRM	0.39	16.41	0.42	26.46	0.36	29.44
	Retand	0.96	40.47	1.40	88.76	1.06	85.47
**LTR-unclassified**		3.39	142.61	5.49	348.91	6.42	519.65
**Other repeats**							
	Pararetrovirus	0.01	0.46	0.22	13.93	0.22	17.71
**DNA transposons**		0.61	25.54	0.71	44.93	0.56	45.12
	TIR/Enspm-CACTA	0.00	0.00	0.00	0.00	0.00	0.00
	TIR/MuDR-Mutator	0.40	16.71	0.49	31.33	0.34	27.15
	TIR/haT	0.18	7.59	0.16	10.00	0.17	13.62
	TIR/PIF-Harbinger	0.03	1.24	0.06	3.60	0.05	4.35
**Tandem repeats**							
	Ribosomal DNA	0.28	11.76	0.40	25.35	0.25	19.97
	Satellite	0.37	15.64	0.62	39.16	0.45	36.71
**Unclassified repeat clusters (GP ≥ 0.01%)**		5.92	249.17	5.64	358.81	5.94	480.84
**Small unclassified clusters (GP < 0.01%)**		13.49	567.21	13.95	886.71	14.10	1141.67
**Total repeats**		85.45	3593.65	85.91	5461.17	85.27	6905.31
**Single copy**		14.55	611.75	14.09	895.83	14.73	1192.53

**Table 2 plants-11-01235-t002:** Statistics for the linear regression analyses carried out between cytotype pairs based on an analysis of the number of reads of Ty1/Copia-like elements, Ty3/Gypsy-like elements, and all repetitive elements (All). (SE: Standard error, Sig.: *** *p*-value < 0.0001).

	8x–6x				6x–4x				8x–4x			
	Slope	SE	R^2^	Sig.	Slope	SE	R^2^	Sig.	Slope	SE	R^2^	Sig.
**Ty1/Copia**	1.37	0.01	0.996	***	1.5	0.01	0.999	***	2.06	0.02	0.996	***
**Ty3/Gypsy**	1.3	0.01	0.999	***	1.5	0	0.999	***	1.95	0.01	0.998	***
**All**	1.31	0	0.997	***	1.5	0	0.999	***	1.97	0.01	0.997	***

## Data Availability

The sequencing datasets presented in this study can be found in online repositories. The names of the repository and bioproject link can be found below: https://www.ncbi.nlm.nih.gov/bioproject/PRJNA802320/ (accessed on 1 April 2022).

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
