# Peer review of "Genome Insights into Autopolyploid Evolution: A Case Study in Senecio doronicum (Asteraceae) from the Southern Alps"

_plants, 2022, doi:10.3390/plants11091235_

Round 1
Reviewer 1 Report
In this manuscript, the authors analyzed the cytotype diversity of S. doronicum from Tête Grosse and the phylogenetic relationship among those three cytotypes based on sequencing data, which provides some evolution insights.
Here are a few comments:
- It's better to show the phenotype of these three polyploids with both figures and descriptions (morphology and living habitat ) in the manuscript.
- It may be interesting to discover any genome variance (beyond ploidy) which may further uncover the evolution of these polyploids.
Author Response
We thank the reviewer for taking the time to review this manuscript and below we enclose our response to his/her comments:
Comment 1: It's better to show the phenotype of these three polyploids with both figures and descriptions (morphology and living habitat ) in the manuscript.
Response: We have created a new figure 1 and added it to the manuscript, in which we illustrate the habitat preferences and morphology in 4x and 8x individuals of Tete Grosse. Unfortunately, only 3 individuals of 6x were found in the population, and their inflorescences were heavily predated by insects, so we did not include them in the figure. Also, we have been cautious when adding any conclusive information regarding habitat preference in the manuscript. At present, this pattern only represents the population studied extensively. It would be too speculative to extrapolate to the whole distribution across the alps.
Comment 2: It may be interesting to discover any genome variance (beyond ploidy) which may further uncover the evolution of these polyploids.
Response: The main point of this study was indeed to provide genomic evidences to study the evolution of polyploids. We believe have done so by adding information based on repetitive DNA in each cytotype. Also we use nuclear markers and plastid genomes to study their evolution. We provide robust evidence of autopolyploid evolution in this species, and infer the potential origin of minority cytotypes. Future studies will be carried out to gain insights across the whole distribution of this species, but these are beyond the scope of this current study.
Reviewer 2 Report
In the Ms entitled ‘Genome insights into autopolyploid evolution: a case study in Senecio doronicum (Asteraceae) from the southern Alps’ authors studied a population-level cytotype screening of 500 individuals from Tête Grosse (Alpes-de-Haute-Provence, France), confirming the co-existence of tetraploid (28.2%) and octoploid cytotypes (71.2%), but also uncovering a small number of hexaploid individuals (0.6%). The analysis of repetitive elements from short read genome skimming data combined with nuclear (ITS) and whole plastid DNA sequences support an autopolyploid origin of the polyploid S. doronicum individuals and provide molecular evidence regarding the sole contribution of tetraploids in the formation of hexaploid individuals.
Authors have done enormous studies from cytotype screening by flow cytometry to illumine sequencing, repetitive DNA analysis, phylogeny, plastid reconstruction etc.
The study is very well designed and nicely executed. The Ms is also very clearly written. The data is enough for publication in the Plants journal.
Author Response
We are happy that the reviewer believes this is an interesting work and are thankful for his/her nice comments regarding the value and execution of this work.